# Self-Supervised Learning Application on COVID-19 Chest X-ray Image Classification Using Masked AutoEncoder

**DOI:** 10.3390/bioengineering10080901

**Published:** 2023-07-29

**Authors:** Xin Xing, Gongbo Liang, Chris Wang, Nathan Jacobs, Ai-Ling Lin

**Affiliations:** 1Department of Computer Science, University of Kentucky, Lexington, KY 40506, USA; 2Department of Radiology, University of Missouri, Columbia, MO 65212, USA; 3Department of Computing and Cyber Security, Texas A&M University-San Antonio, San Antonio, TX 78224, USA; 4Department of Computer Science, University of Missouri, Columbia, MO 65211, USA; 5Department of Computer Science & Engineering, Washington University in St. Louis, St. Louis, MO 63130, USA; 6Department of Biological Sciences, University of Missouri, Columbia, MO 65211, USA; 7Institute for Data Science and Informatics, University of Missouri, Columbia, MO 65211, USA

**Keywords:** vision transformer (ViT), self-supervised learning, chest X-ray image, image classification

## Abstract

The COVID-19 pandemic has underscored the urgent need for rapid and accurate diagnosis facilitated by artificial intelligence (AI), particularly in computer-aided diagnosis using medical imaging. However, this context presents two notable challenges: high diagnostic accuracy demand and limited availability of medical data for training AI models. To address these issues, we proposed the implementation of a Masked AutoEncoder (MAE), an innovative self-supervised learning approach, for classifying 2D Chest X-ray images. Our approach involved performing imaging reconstruction using a Vision Transformer (ViT) model as the feature encoder, paired with a custom-defined decoder. Additionally, we fine-tuned the pretrained ViT encoder using a labeled medical dataset, serving as the backbone. To evaluate our approach, we conducted a comparative analysis of three distinct training methods: training from scratch, transfer learning, and MAE-based training, all employing COVID-19 chest X-ray images. The results demonstrate that MAE-based training produces superior performance, achieving an accuracy of 0.985 and an AUC of 0.9957. We explored the mask ratio influence on MAE and found ratio = 0.4 shows the best performance. Furthermore, we illustrate that MAE exhibits remarkable efficiency when applied to labeled data, delivering comparable performance to utilizing only 30% of the original training dataset. Overall, our findings highlight the significant performance enhancement achieved by using MAE, particularly when working with limited datasets. This approach holds profound implications for future disease diagnosis, especially in scenarios where imaging information is scarce.

## 1. Introduction

The COVID-19 pandemic has brought attention to the vital role of artificial intelligence (AI) in combating infectious diseases, specifically through the analysis of lung images, such as X-ray chest images. Computer-aided diagnosis (CAD) has emerged as a promising tool for accurate and rapid diagnosis in this context. deep learning (DL) models, known for their exceptional performance in computer vision tasks such as image recognition [1,2,3,4], semantic segmentation [5,6], and object detection [7,8,9], have increasingly been adopted for CAD and other healthcare applications [10,11,12,13].

Despite the significant potential of DL models in medical data analysis, there are several practical challenges impeding their widespread adoption. First, medical datasets are often smaller compared to those used for natural image analysis, such as the widely used ImageNet dataset [14]. DL models have numerous parameters and require substantial data for effective training. Consequently, training such models with limited datasets can be challenging and may lead to overfitting [15,16]. Second, labeling medical data is a resource-intensive and time-consuming process. CAD relies on labeled data for training, but labeling medical data necessitates specialized medical knowledge and expertise, making it more demanding than labeling natural images. Lastly, current DL models for medical image analysis primarily rely on convolutional neural networks (CNNs) [17,18]. While CNNs excel at capturing local features, they may not be optimally suited for capturing global information across an entire image. Therefore, further research is necessary to develop more effective DL network architectures capable of capturing both local and global features for medical image analysis, particularly when dealing with limited datasets.

To address these challenges, various methods have been proposed. For example, transfer learning [19] has gained widespread usage, wherein DL models are pretrained on large-scale natural image datasets such as ImageNet and subsequently fine-tuned on smaller medical datasets. This approach helps mitigate the overfitting issue caused by limited medical datasets, although it may not fully bridge the gap between natural and medical images. Regarding labeling issues, weakly supervised learning (WSL) [20,21] has become popular, where models are trained using only image-level labels for tasks such as object detection [22]. However, WSL may not be suitable for classification tasks that still require image-level labels. Recently, novel DL models such as the Vision Transformer (ViT) [23] and its variants [24,25] have demonstrated promising results in capturing global information from medical images. Nevertheless, these models often necessitate extensive amounts of data for effective training.

In our study, we explored a novel method for medical image analysis that addresses the challenges associated with training strategy and limited medical datasets. We propose the utilization of self-supervised learning (SSL) [26], a method that leverages the intrinsic attributes of the data as pretraining tasks, eliminating the reliance on labeled data. SSL implementation involves utilizing attributes such as image rotation prediction [27], patch localization [28], and image reconstruction [29], which can be accessed without manual labeling. Additionally, by substituting the conventional CNN backbone with a Vision Transformer (ViT) model, our method effectively captures both local and global features of medical images. To accomplish this, we employ a Masked Autoencoder (MAE) model [30], a self-supervised learning approach that utilizes the ViT model as its backbone. By combining SSL with ViT through the MAE model, we anticipate that this method can contribute to more accurate and efficient medical image analysis.

To our best knowledge, we are the first to apply MAE on COVID-19 X-ray imaging. During the exploration, we found an innovative application of MAE on the limited dataset, which is not studied by the previous work [30]. We demonstrated the superior performance of the MAE model compared to baseline models, explored the influence of mask ratios on the MAE model’s performance, and evaluated the MAE model’s performance using different proportions of limited training data. The contributions of our study are as follows:We conducted a comparative analysis of various training strategies using the same public COVID-19 dataset and observed that the MAE model outperformed other approaches, demonstrating superior performance.To further investigate the impact of different mask ratios on the MAE model’s performance, we examined how varying mask ratios affected the effectiveness of the model. Our experiments revealed that the model achieved its best performance with a mask ratio of 0.4.Through extensive evaluations, we examined the applicability of the MAE model across different proportions of available training data. Remarkably, the MAE model achieved comparable performance even when trained with only 30% of the available data.

## 2. Materials and Methods

### 2.1. Data

Our study adopted the chest X-ray classification dataset: COVIDxCXR-3 [31], which is a public dataset with more than 29,000 chest X-ray images, for positive/negative detection. COVIDxCXR-3 collects the data from different public data sources: covid-chest x-ray-dataset [32], Figure 1 COVID-19 Chest X-ray Dataset Initiative [33], Actualmed COVID-19 Chest X-ray Dataset Initiative [34], COVID-19 Radiography Database—Version 3 [35], RSNA Pneumonia Detection Challenge [36], RSNA International COVID-19 Open Radiology Database (RICORD) [37], BIMCV-COVID19+: a large annotated dataset of RX and CT images of COVID-19 patients [38], and Stony Brook University COVID-19 Positive Cases (COVID-19-NY-SBU) [39]. Figure 1 visualizes the positive/negative image samples of the COVID-19 subject. Table 1 shows the details of the COVIDxCXR-3 dataset distribution. The dataset has a multinational cohort of over 16,600 patients. The whole dataset is split into training and testing sets by the dataset authors. The training dataset has 13,992 negative and 16,490 positive images. The testing dataset has 200 negative and positive images, respectively.

During the implementation of our experiment, we configured the image size to 224 × 224 pixels, while normalizing the image pixel values within the range of 0 to 255. Under normal circumstances, in order to provide an unbiased evaluation of a model, cross-validation is typically conducted during the training process. However, as demonstrated in Table 1, there exists an inequality in both image and patient distributions. There are 15,994 positive images for COVID-19, derived from 2808 COVID-19-positive patients, indicating multiple X-ray images per patient in the dataset (approximately 6 images per patient). Meanwhile, the dataset authors have not provided the specific subject information, precluding the possibility of conducting subject-level cross-validation. Concurrently, cross-validation at the image level would result in data leakage. Therefore, we chose to adopt the train/test split as defined by the dataset authors in these particular circumstances.

### 2.2. Vision Transformer

Since ViT is built upon the self-attention mechanism and many works adopt multi-head attention in the implementation, we first introduce the basics of the attention mechanism, then describe the ViT architecture.

The attention mechanism includes three inputs: a query (*Q*), a key (*K*), and a value (*V*). The attention operation is defined as Equation (Equation 1):(1)$$\mathrm{Attention}\left(Q,K,V\right)=\mathrm{Softmax}\left(\frac{QK^{T}}{\sqrt{d_{k}}}\right)V$$

Afterward, the multi-head attention is defined as Equation (Equation 2):(2)$$\begin{array}{cc} \; & \mathrm{MultiHeadAttn}\left(Q,K,V\right)=\mathrm{Concat}\left(head_{1},...,head_{n}\right)W^{o}\\ \; & head_{i}=\mathrm{Attention}\left(QW_{i}^{Q},KW_{i}^{K},VW_{i}^{V}\right) \end{array}$$
where $W_{i}^{Q}$, $W_{i}^{K}$, and $W_{i}^{V}$ are learnable projection matrices.

Figure 2 illustrates the ViT structure, which has a patch embedding module and N× stacked transformer encoder blocks. Each transformer encoder block contains a multi-head attention (MSA), two layer normalization (LN) [40], and a multi-layer perceptron (MLP).

In the implementation, the input image *I* is first transformed into a series of patch embeddings: (3)$$z_{0}=\mathrm{PatchEmbedding}\left(I\right)$$

The patch embeddings are forwarded through the transformer encoder under the following operations:(4)$$z^{l}=MLP\left(Norm\left(MSA\left(Norm\left(z^{l-1}\right)\right)\right)\right)$$
where $z^{l-1}$ and $z^{l}$ are the *l*-th transformer encoder block input and output, respectively.

### 2.3. MAE

Originally, MAE is a proficient self-supervised learning method employed in Natural Language Processing (NLP) tasks. Building upon its initial foundations, the MAE has significantly broadened its application scope beyond the NLP field, marking its presence in the field of computer vision. This widening of its application is facilitated by the universality of its core principle—the strategic “masking” operation. This operation forms the core of the self-supervised learning methodology, by selectively omitting sections of input data to create a challenge for the model to reconstruct these masked elements. This process allows the model to develop a robust understanding of the intrinsic structure and properties of the target dataset, optimizing its ability to generate representative features during the pretraining phase. This initial phase is followed by a fine-tuning stage, which utilizes labeled input data to further refine the model’s comprehension of the dataset, thereby improving its overall performance. This two-tiered approach equips the model with the essential tools to tackle novel data and perform reliably on the target dataset. The successful adaptation of the MAE methodology to the realm of computer vision was achieved by employing techniques parallel to its NLP counterpart. Images are decomposed into a multitude of patches, a subset of which are randomly masked. Subsequently, the model is trained to perform the reconstruction pretraining task, effectively learning to predict the obscured sections of the image. Following this, a fine-tuning phase is undertaken with labeled data, ensuring the model’s efficient performance on the target dataset.

When considering the choice of backbone for the MAE model, ViT emerges as an optimal option when compared to the convolutional neural network (CNN). As previously mentioned, the ViT architecture offers distinct advantages. One notable feature of the ViT model is its initial operation, where the input image is segmented into various patches. This patch-based approach can easily utilize the masking of random regions. Given these advantages, it is judicious to adopt a ViT architecture as the backbone for the MAE model.

Figure 3 illustrates the comprehensive workflow of a self-supervised learning system based on the MAE, encompassing two essential stages: the pretext task of image reconstruction and the subsequent fine-tuning stage. When it comes to the architecture of the model during the pretraining stage, it incorporates a Vision Transformer (ViT) as both the encoder and decoder. Serving as an encoder, the ViT is applied to mask certain segments of the input image patches. In its role as a decoder, it is tasked with the restoration of the masked patches. Upon transitioning to the fine-tuning phase, the pretrained ViT encoder is trained further with samples and labels from the target dataset. In the context of our implementation, we elected to employ a ViT-small structure with a hidden size of 768 as the encoder and the standard decoder within the framework of the MAE. The overall pipeline, founded on block representation, is presented in Figure 4.

### 2.4. Loss Function

Since our work concentrates on binary classification, the overall loss function is a binary cross-entropy. For a chest X-ray image *V* with label *l* and probability prediction p(l|V), the loss function is
(5)loss(l,V)=llog(p(l|V))+(1−l)log(1−p(l|V))
where the label l=0 indicates a negative sample and l=1 indicates a positive sample, respectively.

### 2.5. Implementation and Metrics

We implemented the experiment models using PyTorch. We trained and tested the models based on the default setting of the dataset. For the pretrained baseline, the model is pretrained on ImageNet [14]. For the model training, we set the batch size to 16. Adam optimizer [41] with beta1=0.9, beta2=0.999, and a learning rate of $1\times 10^{-4}$ was used during the training. For the SSL model, we pretrained the model with 100 epochs. In the fine-tuning stage, we trained all the models for 40 epochs.

To evaluate the performance of our model, we used accuracy (Acc), area under the curve of receiver operating characteristics (AUC), F1 score (F1), Precision, Recall, and Average Precision (AP) as our evaluation metrics. We evaluated the training computation cost by the average epoch training time (e-Time). The accuracy is calculated with the following Equation (Equation 6):(6)$$\mathrm{Accuracy}=\frac{\mathrm{TP}+\mathrm{TN}}{\mathrm{TP}+\mathrm{TN}+\mathrm{FP}+\mathrm{FN}}$$
where TP is the True Positive, TN is the True Negative, FP is the False Positive, and FN is the False Negative.

The precision is calculated by the following Equation (Equation 7):(7)$$\mathrm{precision}=\frac{\mathrm{TP}}{\mathrm{TP}+\mathrm{FP}}$$

The recall is calculated by the following Equation (Equation 8):(8)$$\mathrm{recall}=\frac{\mathrm{TP}}{\mathrm{TP}+\mathrm{FN}}$$

The F1-score is calculated by the following Equation (Equation 9):(9)$$F1-\mathrm{score}=2\times \frac{\mathrm{precision}\cdot \mathrm{recall}}{\mathrm{precision}+\mathrm{recall}}$$

AUC curves compare the true positive rate and the false positive rate at different decision thresholds. AP summarizes a precision–recall curve as the weighted mean of precision achieved at each threshold.

## 3. Results

### 3.1. Model Performance Increasing by MAE

We conducted training experiments on the ViT model architecture using three different approaches: ViT-scratch, ViT-pretrain, and ViT-MAE. In the ViT-scratch approach, the ViT model was trained directly on the medical image data. The ViT-pretrain approach involved fine-tuning a pretrained ViT model on ImageNet using the medical image data. ViT-MAE refers to training the ViT model using the MAE pipeline. Accuracy was chosen as the performance metric. As depicted in Table 2, ViT-MAE achieved a remarkable accuracy of 0.985 in COVID-19 positive/negative detection, surpassing the other approaches (ViT-scratch accuracy = 0.7075 and ViT-pretrain accuracy = 0.9350) on the same dataset. To further compare ViT-MAE with CNN models, namely, ResNet50 and DenseNet121, we conducted additional experiments. It was observed that ViT-MAE outperformed both ResNet50 and DenseNet121 in terms of all metrics, except for AUC, where the difference was minimal. We think this minimal difference is due to (1) the model experiment’s randomness, a common characteristic of machine learning models, and (2) the size of the test dataset. The size of the test dataset would overestimate/underestimate the model. In our experiments, compared with the training dataset, the test dataset is relatively small with only 400 images, which may overestimate the models. However, in terms of the same ViT backbone, we think ViT-MAE models exhibit comparable performance in our experiments. Figure 5 illustrates the AUC curves for the three training approaches, clearly demonstrating that ViT-MAE outperforms the other strategies in terms of AUC performance.

We studied statistical tests that compare the ViT-MAE performance with ViT-scratch and ViT-pretrain. The metric chosen to evaluate their performance was accuracy. To ensure robustness, we conducted four independent experiments for each ViT model, employing different random seeds. The statistical summary of the three pretraining methods yielded the following mean and standard deviation values: For ViT-scratch, the mean was 0.7135 with a standard deviation of 0.0142. For ViT-pretrain, the mean was 0.9293 with a standard deviation of 0.0207. Lastly, for ViT-MAE, the mean was 0.9775 with a standard deviation of 0.006. In order to assess the significance of the differences in performance, we conducted two *t*-tests: ViT-scratch vs. ViT-MAE and ViT-pretrain vs. ViT-MAE. The resulting *p*-values for the two group *t*-tests were found to be less than 0.001 and 0.02, respectively. Our analysis revealed that ViT-MAE significantly outperformed ViT-scratch, indicating the critical influence of the training strategy on model performance. Additionally, we observed a relatively narrow performance gap between ViT-MAE and ViT-pretrain. These findings suggest that while ViT-MAE exhibits superior performance compared to ViT-scratch, the disparity in performance between ViT-MAE and ViT-pretrain is comparatively smaller.

### 3.2. Mask Ratio Influence on MAE Performance

Since the pretraining of ViT-MAE is a reconstruction task, the mask ratio of the input image is a parameter that may affect the final performance. In this section, we study the mask ratio influence on ViT-MAE training. Table 3 shows the performance of different mask ratios over the MAE pretraining stage. Figure 6 illustrates the AUC curves of different mask ratios. The 40% percentage mask ratio outperforms the other mask ratio situations with Acc = 0.9850 and AUC = 0.9957. The mask ratio result indicates that a large mask ratio may decrease the final performance for the medical image dataset, while the large mask ratio (mask ratio = 0.75) shows good performance in the natural dataset. We think this may due to the difference between the medical and natural datasets, and the reconstruction results on the medical image may not show better performance than the natural image. We prove our thoughts in Section 3.4.

### 3.3. MAE Performance on the Limited Training Dataset

One advantage of self-supervised learning is that people can use a small labeled dataset to train a large DL model well. To explore the potential of SSL, we conduct experiments on the limited training dataset. We randomly split the partial training dataset to train our model from 10% to 90%. Table 4 shows the performance under different percentage splitting, and Figure 7 shows the AUC curves of different percentage situations. It appears that using only 30% of the training dataset is sufficient to achieve better performance than that of the ViT-pretrain scenario with the whole training dataset (94.25 vs. 93.5). Meanwhile, it is clear that increasing the labeled training percentage will contribute to better performance of the model. The promising results provide the potential training procedure for small medical dataset training on DL models.

### 3.4. Visualization of MAE on Image Reconstruction

We present a visualization of the X-ray image reconstruction phase. As demonstrated in Figure 8, the depiction includes both Covid and Normal subjects, showcasing the original, masked, and reconstructed images. Upon visual comparison, it can be observed that the reconstructed images are relatively coarse. However, it should be emphasized that our primary objective is not to achieve pixel-perfect image reconstruction but to ensure that the deep learning model’s parameters are properly initialized for the fine-tuning process on the specific dataset. Concurrently, the rough outcome of the reconstruction task implies that increasing the mask ratio will not contribute to enhanced model performance during the fine-tuning stage. This is due to the model’s inability to extract additional learning during the reconstruction stage.

## 4. Discussion

Our study involved an in-depth exploration of the MAE through a comprehensive series of experiments utilizing a publicly available COVID-19 Chest X-ray image dataset. Our study is innovative as we applied MAE to an X-ray imaging dataset for COVID-19 diagnosis, which has not been reported before. Further, we demonstrated that MAE exhibits remarkable efficiency when applied to labeled data, delivering comparable performance to utilizing only 30% of the original training dataset. The findings may have profound implications for the diagnosis of various diseases with the limited imaging dataset in the future, given that we showed that the accuracy can be maintained even with a reduced, smaller dataset. Our findings also yield several significant insights on the application of MAE in medical imaging as follows: First, by leveraging self-supervised learning with MAE, we observed notable improvements in model performance compared to alternative training methods. This underscores the efficacy of MAE in the context of medical image analysis. Second, the performance of the MAE model on medical images was found to be influenced by the masked ratio employed during training. Notably, we achieved optimal results with a masked ratio of 0.4 in our implementation. This indicates the importance of carefully selecting the appropriate ratio to achieve the best performance. Finally, our study demonstrates that MAE operates as a labeled data-efficient model, showcasing comparable performance even when trained on a partial dataset. This finding highlights the potential of MAE in situations where acquiring large quantities of labeled data may be challenging or resource-intensive.

In our implementation, we utilized a Vision Transformer (ViT) as the DL model. Compared to traditional CNN models, the ViT has shown promising performance across a range of tasks, but it is prone to being data-hungry during the training phase. We conducted experiments on the same DL model and training setting but with different training strategies: ViT-scratch, ViT-pretrain, and ViT-MAE. The results demonstrate the efficacy of self-supervised learning, yielding an accuracy of 0.985 and an AUC of 0.9957. Meanwhile, we compared the performance of the MAE model with the CNN-based models [42], which shows MAE outperforms the CNN models.

In our experiments, we explored the association between the mask ratio, a hyperparameter in the masked token reconstruction task, and the model’s performance. We set the mask ratio from 0.4 to 0.8 and found that increasing the mask ratio led to a decrease in performance, which is different from the mask ratio result (0.75) of MAE on natural images [30]. To explain this trend, we visualized the original images, masked images, and reconstructed images. Comparing the original and reconstructed images, we observed that the reconstructed images were blurrier. The goal of reconstruction pretraining is to initialize the model parameters and enhance the model’s understanding of the medical dataset. However, a high mask ratio may hinder the reconstruction process and weaken the model’s understanding ability. Therefore, the mask ratio is a crucial factor in practical implementation.

To further highlight the advantages of self-supervised learning, we conducted limited dataset experiments. We randomly sampled the training dataset from 10% to 90% and applied the sample to conduct the reconstruction pretraining and fine-tuning of the model. The results strongly indicate the advantage of self-supervised learning on limited data. For example, using only a 30% sample of the training dataset, the ViT-MAE model still achieved 0.9425 accuracy, comparable to the performance of the ViT-pretrain model using the entire training dataset. This is particularly important in clinical applications, where datasets are often limited. Training large DL models on limited data can be challenging and can easily lead to overfitting due to the large number of parameters in the model. By this pretraining stage, the model can learn the good representation of the target dataset. Compared to the pretained model by natural image datasets, such as the ImageNet dataset, the pretraining stage MAE model has a narrow gap for the target dataset (i.e., small medical dataset) and is suitable for the later fine-tuning stage. Therefore, the MAE is suitable for the limited dataset. Additionally, using a smaller training dataset to train a large DL model reduces the cost of labeling, as traditional supervised DL training requires a large labeled training dataset to ensure model convergence. However, data labeling can be another issue when the dataset size is large, as in the case of the ImageNet dataset with one million images. Furthermore, medical image labeling often requires professional domain knowledge, such as an X-ray radiologist, to ensure accurate labeling.

The limitations of our study include the focus on COVID-19 alone. For future work, we plan to extend our work in two directions: first, we will extend the MAE model to handle 3D medical images, such as 3D brain imaging for Alzheimer’s Disease [43,44,45]; second, we will explore the potential of the MAE for other tasks, such as image segmentation and localization [6,46], beyond image classification.

In conclusion, we applied MAE to the X-ray imaging dataset for COVID-19 diagnosis and illustrated that MAE exhibits remarkable efficiency when applied to labeled data, delivering comparable performance to utilizing only 30% of the original training dataset. Overall, our findings highlight the significant performance enhancement achieved by using MAE, particularly when working with limited datasets. This approach holds profound implications for future disease diagnosis, especially in scenarios where imaging information is scarce.

## Figures and Tables

**Figure 1 bioengineering-10-00901-f001:**
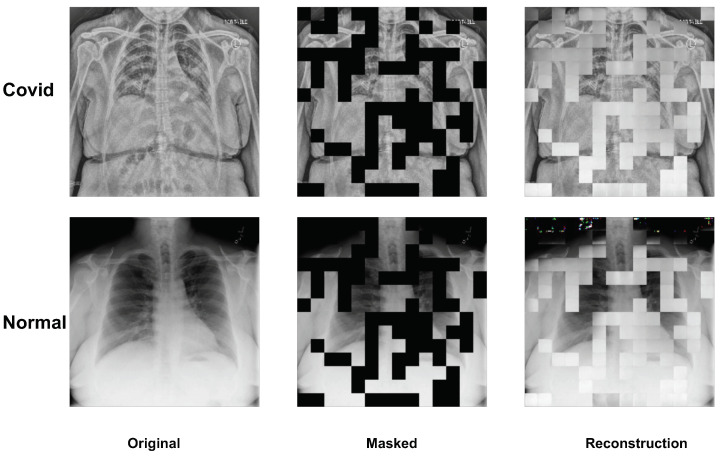
The visualization of the chest X-ray image of the COVIDxCXR-3. The first row shows the negative subjects and the second row shows the positive subjects. The input image size is 224 × 224. We normalize the image pixel from 0 to 255.

**Figure 2 bioengineering-10-00901-f002:**
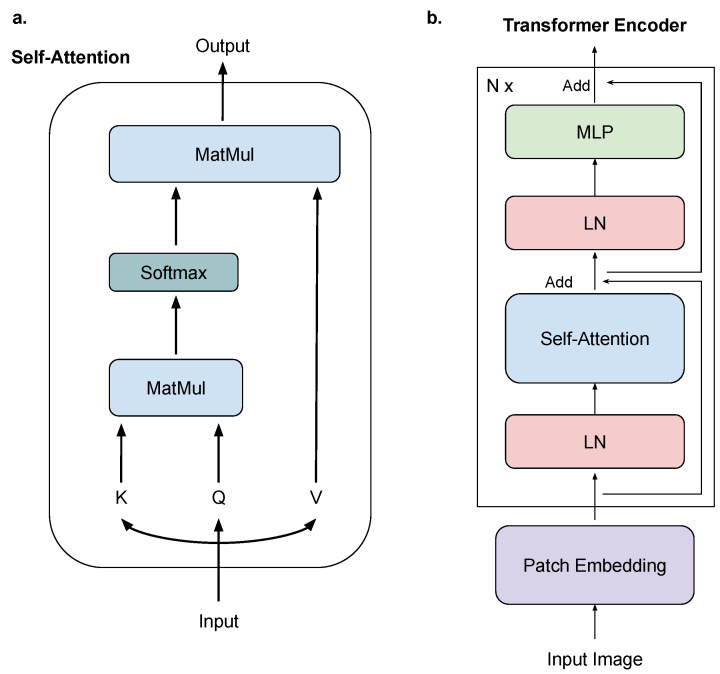
The structure of Vision Transformer. Sub-figure (**a**) illustrates the structure of the self-attention module. Sub-figure (**b**) shows the architecture of the Vision Transformer encoder.

**Figure 3 bioengineering-10-00901-f003:**
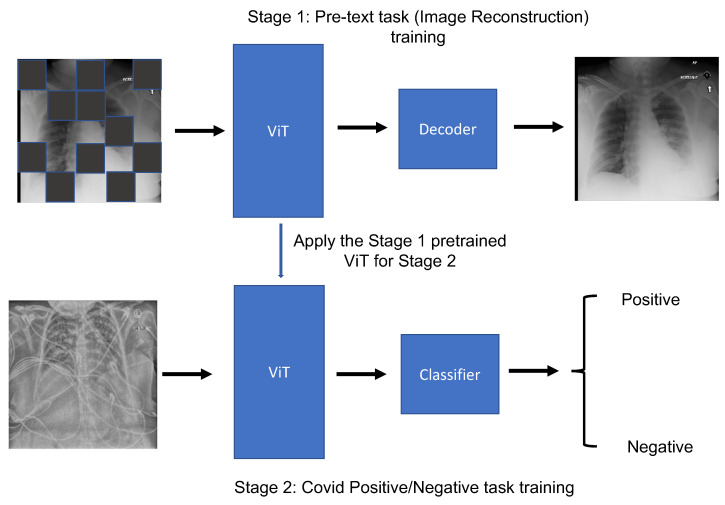
The workflow of the MAE method on the COVID-19 classification task. There are two stages for MAE training. The first stage is the image reconstruction pretraining stage, with the ViT backbone as the image encoder. The second stage is a fine-tuning stage, with the ViT backbone as the feature extractor for the labeled images.

**Figure 4 bioengineering-10-00901-f004:**
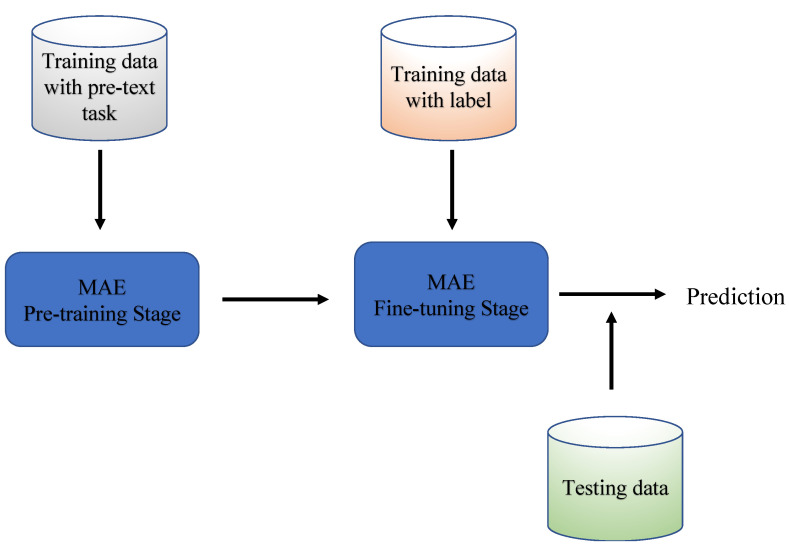
The block presentation of the MAE pipeline.

**Figure 5 bioengineering-10-00901-f005:**
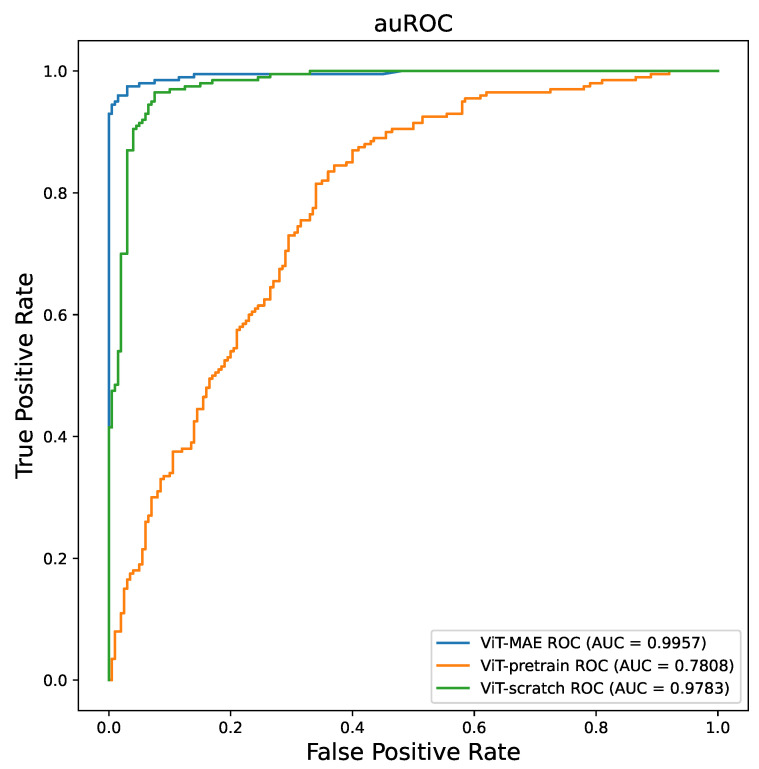
The AUC plot of the different training strategies for ViT model.

**Figure 6 bioengineering-10-00901-f006:**
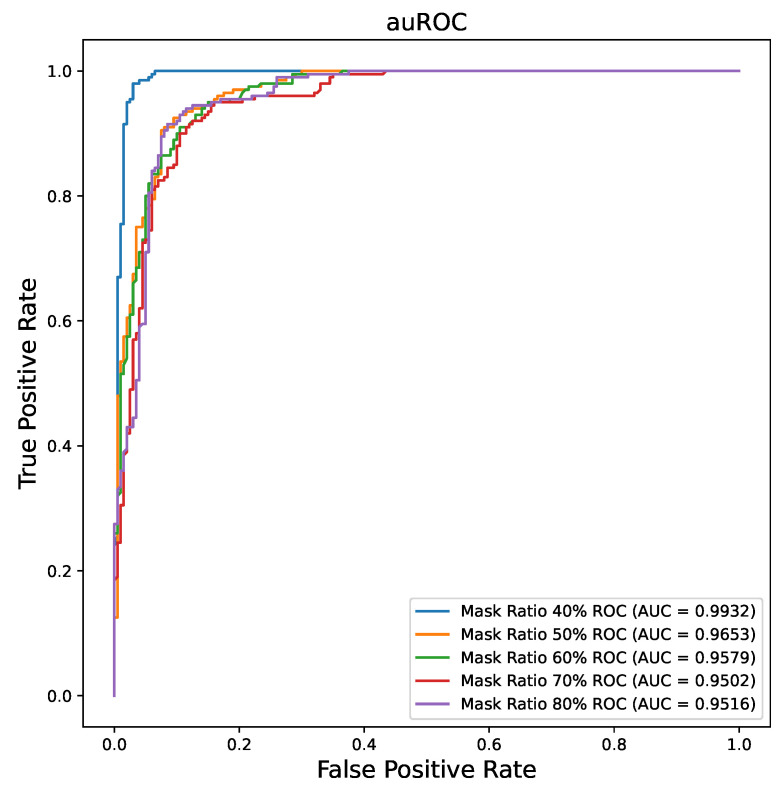
The AUC plot of the different mask ratios for ViT-MAE pretraining.

**Figure 7 bioengineering-10-00901-f007:**
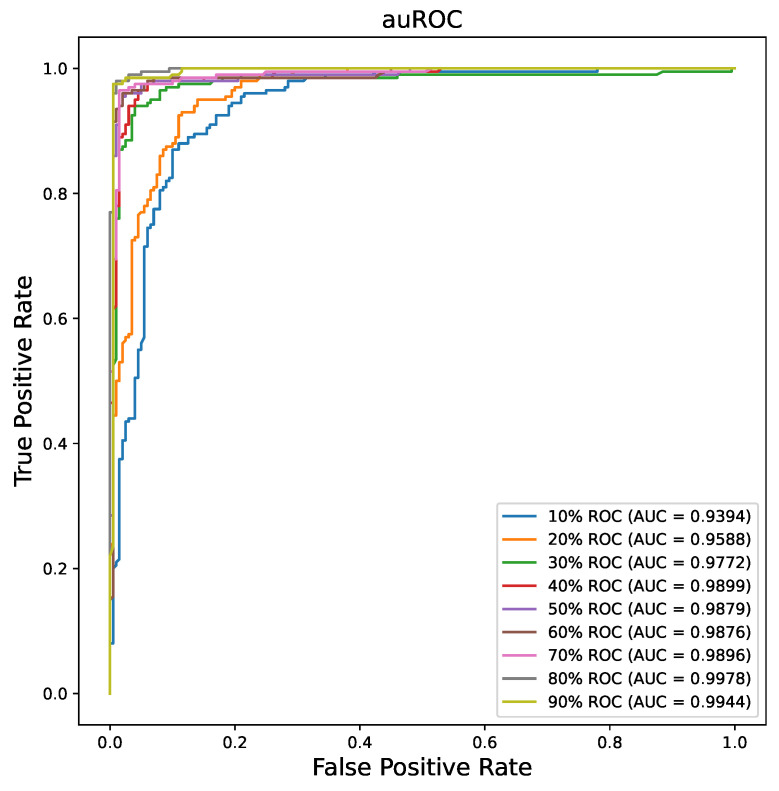
The AUC plot of the different percentages of training dataset for ViT-MAE model.

**Figure 8 bioengineering-10-00901-f008:**
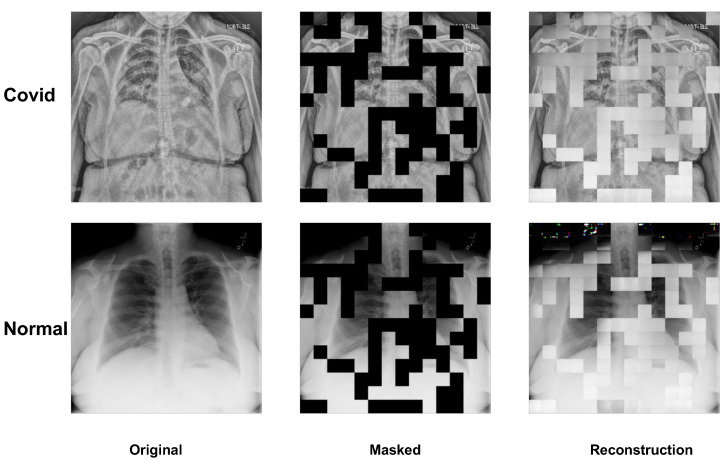
The visualization of the MAE for the image reconstruction pretraining. From the left column to the right are the original input image, the random masked image, and the reconstruction image. Even though the final reconstruction is not well-defined, the target of the pretraining stage is boosting the initial parameters of the ViT model.

**Table 1 bioengineering-10-00901-t001:** The images and patients distribution of the dataset COVIDxCXR-3.

Type	Negative	Positive	Total
	Images Distribution		
Train	13,992	15,994	29,986
Test	200	200	400
	Patients Distribution		
Train	13,850	2808	16,648
Test	200	178	378

**Table 2 bioengineering-10-00901-t002:** The performance of different training strategies over the ViT model. ViT-MAE model outperforms the other two training strategies.

Type	Acc	AUC	F1	Precision	Recall	AP
DenseNet121	0.9775	**0.9970**	0.9771	0.9948	0.96	0.9750
ResNet50	0.9650	0.9969	0.9641	0.9894	0.94	0.9601
ViT-scratch	0.7075	0.7808	0.7082	0.7065	0.7100	0.6466
ViT-pretrain	0.9350	0.9783	0.9340	0.9484	0.9200	0.9125
ViT-MAE	**0.9850**	0.9957	**0.9850**	**0.9950**	**0.9850**	**0.9859**

**Table 3 bioengineering-10-00901-t003:** The performance of different mask ratios over the MAE pretraining stage. The pre-training mask ratio = 0.4 of MAE outperforms the other pretraining strategies.

Ratio	Acc	AUC	F1	Precision	Recall	AP
0.4	0.9850	0.9957	0.9850	0.9850	0.9850	0.9559
0.5	0.9100	0.9653	0.9086	0.9277	0.8950	0.8783
0.6	0.8875	0.9579	0.8819	0.9282	0.8400	0.8597
0.7	0.8900	0.9502	0.8894	0.8939	0.8850	0.8486
0.8	0.8925	0.9516	0.8900	0.9110	0.8700	0.8576

**Table 4 bioengineering-10-00901-t004:** The performance of different percentages of training datasets at the MAE pretraining stage. The pretraining mask ratio = 0.4 of MAE outperforms the other pretraining strategies.

Percentage (%)	Acc	AUC	F1	Precision	Recall	AP
10	0.8800	0.9394	0.8776	0.8958	0.8600	0.8404
20	0.8925	0.9588	0.8877	0.9290	0.8500	0.8646
30	0.9425	0.9772	0.9415	0.9585	0.9250	0.9242
40	0.9600	0.9866	0.9602	0.9554	0.9650	0.9395
50	0.9675	0.9879	0.9673	0.9746	0.9600	0.9556
60	0.9650	0.9876	0.9645	0.9794	0.9500	0.9554
70	0.9675	0.9896	0.9669	0.9845	0.9500	0.9602
80	0.9775	0.9978	0.9771	0.9948	0.9600	0.9750
90	0.9825	0.9944	0.9823	0.9949	0.9700	0.9800

## Data Availability

Data used in this article are from the dataset: COVIDxCXR-3. The collected data are available here: https://www.kaggle.com/datasets/andyczhao/covidx-cxr2 (2 June 2022).

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
