# Peer review of "Self-Supervised Learning Application on COVID-19 Chest X-ray Image Classification Using Masked AutoEncoder"

_bioengineering, 2023, doi:10.3390/bioengineering10080901_

Round 1

Reviewer 1 Report

1. How does the proposed work overcome the problem of model overfitting?

2. The authors are suggested to conduct cross-validation in the training process rather than the typical train/test split

3. Statistical tests should be conducted when comparing the performance among methods/models to see significant differences.

4. The paper fails to provide a thorough analysis of the proposed technique. The authors do not provide any detailed explanation of the technique or its underlying algorithm. As a result, it is challenging for readers to understand the proposed technique, and its potential limitations or drawbacks are not adequately addressed. I suggest the authors read studies performed by scholars such as Ranjbarzadeh et al., Jafarzadeh et al., and their groups

5. Authors need to provide a block representation for the proposed method for better understanding by the reader.

The following papers are good examples:

https://doi.org/10.1155/2022/5052435

https://doi.org/10.1007/s12559-022-10072-w

Author Response

We thank the reviewer’s thoughtful comments. We have revised the manuscript accordingly. Please see blow the point-by-point responses.

  1. How does the proposed work overcome the problem of model overfitting?

Response: We appreciate the reviewer pointing this out as overfitting could be an issue using the limited training dataset from the COVID-19 medical imaging in the present study. We addressed the issue using MAE (Masked AutoEncoder), a method used in self-supervised learning (SSL). In the SSL model, the initial pre-training phase involves training the model on certain image attributes without labels, such as image recovery or predicting image rotation. This pre-training allows the model to acquire a strong representation of the target dataset. Unlike models pre-trained on natural image datasets like ImageNet, the SSL model's pre-trained parameters closely align with the target dataset (e.g., a small medical dataset), making them suitable for subsequent fine-tuning stages. Consequently, MAE proves to be effective for dealing with limited datasets. To demonstrate the capabilities of MAE, we conducted experiments using a limited training dataset, and the results revealed that the MAE method performs well even when trained on a small portion of the dataset. We added the MAE contribution on the overfitting solution in the Discussion section (page 12, lines 290-297).

  1. The authors are suggested to conduct cross-validation in the training process rather than the typical train/test split.

Response: We agree with the reviewer that in the training process, conducting cross-validation is a crucial step to ensure a fair evaluation of model performance. However, by examining Table 1 in our paper, it becomes apparent that the distribution of images and patients is uneven. Specifically, there are 15,994 Covid-19 positive images but only 2,808 Covid-19 positive patients, indicating that each patient has multiple X-ray images within the dataset (approximately 6 images per patient). Complicating matters, the dataset authors did not provide subject information, making it very challenging to perform cross-validation at the subject level. Conducting image-level cross-validation would lead to data leakage, which is undesirable. Given these circumstances, we were unable to conduct experiments using cross-validation and have explicitly addressed this limitation in Section 2.1 for clarity (pages 3-4, lines 99-109).

  1. Statistical tests should be conducted when comparing the performance among methods/models to see significant differences.

Response: We appreciate the reviewer’s suggestion and added the experiments with multiple random seeds. The detailed statistical test results of different ViT training strategies can be found in Section 3.4 (page 8, lines 195-209).

  1. The paper fails to provide a thorough analysis of the proposed technique. The authors do not provide any detailed explanation of the technique or its underlying algorithm. As a result, it is challenging for readers to understand the proposed technique, and its potential limitations or drawbacks are not adequately addressed. I suggest the authors read studies performed by scholars such as Ranjbarzadeh et al., Jafarzadeh et al., and their groups.

Response: We appreciate the reviewer’s suggestion. We have read the suggested studies (Refs 9 and 10) and revised the method in Section 2.3 (Pages 5-6, lines 120-144).

  1. Authors need to provide a block representation for the proposed method for better understanding by the reader.

The following papers are good examples:

https://doi.org/10.1155/2022/5052435

https://doi.org/10.1007/s12559-022-10072-w

Response: We appreciate the reviewer’s suggestion. We provided a block representation for our method (Section 2.3, Figure 4). We also included the two suggested references in our revision (Refs 9 and 10).

Reviewer 2 Report

By comparing different training strategies, the authors determined the optimal training method MAE under specific lung image datasets. Meanwhile, the performance of MAE was compared again according to different mask ratios to determine the optimal mask ratio and the superior performance of MAE on small sample data. However, the innovation of this article is slightly insufficient, and the authors are advised to transfer to other journals.

1. The authors seem not to be innovative in the method, but just used someone else's method to find the most appropriate value on the specific dataset. If you do have a unique innovation, you are advised to highlight it.

2. The content of Table 2 seems not intuitive enough. It is suggested that the authors make the best value bold or underline, so as to clearly observe the superiority of your method.

3. In the comparison results in Table 2, your AUC result seems not to be better than other methods. It’s suggested authors add the reasons and improvements for this result.

4. As the authors said the number of dataset your method focuses on is too small, but this seems to be a problem can be solved, you just need to validate your method on multiple data sets. It’s suggested to add relevant content.

5. The authors described your workflow, but did not explain why you can get better results with a visual converter (ViT) model instead of a traditional CNN backbone. In other words, the authors only described how the model works, but not why it works.

Need to be modified.

Author Response

We thank the reviewer’s thoughtful comments. Please see blow the point-by-point responses. We have also revised the manuscript accordingly. 

  1. The authors seem not to be innovative in the method, but just used someone else's method to find the most appropriate value on the specific dataset. If you do have a unique innovation, you are advised to highlight it.

 Response: We thank the reviewer for the comment and apologize that we did not make it clear in the previous version. The innovation of the study is twofold. First, we applied MAE (Masked AutoEncoder), a method used in self-supervised learning (SSL), to the X-ray imaging dataset for COVID-19 diagnosis, which has not been reported before. Second, we demonstrated that MAE exhibits remarkable efficiency when applied to labeled data, delivering comparable performance to utilizing only 30% of the original training dataset. The finding is not only innovative but may also have profound implications for various diseases diagnosis with limited imaging dataset in the future, given that we showed that the accuracy can be maintained even with various reduced, smaller dataset. We added this innovation application of MAE in our introduction section with the contribution of our work and the discussion section. (page 2, lines 67-69; page 12, lines 247-253).

  1. The content of Table 2 seems not intuitive enough. It is suggested that the authors make the best value bold or underline, so as to clearly observe the superiority of your method.

Response: Thank you for reviewer’s suggestion. We made the best value bold to make sure the superiority of our method (Table 2).

  1. In the comparison results in Table 2, your AUC result seems not to be better than other methods. It’s suggested authors add the reasons and improvements for this result.

Response: We appreciate the reviewer’s insightful comments. Observing the minimal AUC disparity between ViT-MAE (AUC=0.9957) and DenseNet121 (AUC=0.9970), we think the AUC performance of ViT-MAE is essentially comparable. Furthermore, it’s important to remember that AUC isn’t the only measure of model performance. Our proposed model demonstrates superiority over baseline models in other metrics, proving its effectiveness. The potential reasons for the observed AUC performance could be 1) inherent randomness, a common characteristic of machine learning models, and 2) the size of the test dataset. The size of the test dataset would overestimate/underestimate the model. In our experiments, compared with the training dataset, the test dataset is relatively small with only 400 images, which may overestimate the models. Therefore, the performance difference between the proposed model and the baseline model might not be significant. We have provided further discussion regarding the AUC in section 3.1 (pages 7-8, lines 185-191) of our manuscript.

  1. As the authors said the number of dataset your method focuses on is too small, but this seems to be a problem can be solved, you just need to validate your method on multiple data sets. It’s suggested to add relevant content.

Response: We apologize for the confusion in the original manuscript. The COVID-19 lung imaging database we used in the study was quite large. The COVIDxCXR-3 dataset that we used includes multiple public datasets for Covid-19 Chest X-ray imaging, such as the BIMCV-COVID19+ dataset and the Covid-19 radiography database. However, not every disease could have such a large wealth of imaging data for training and testing, or for validation. In conditions like this, it will be very helpful to have a method that can deal with the small dataset while maintaining comparable accuracy. The purpose of the study is to address the gap. We conducted experiments using a large dataset but used different percentages (ranging from 10% to 90%) of limited training data to assess the applicability of the model. The results from these experiments indicate that the model consistently demonstrates comparable performance even when trained with small datasets. Through the experiments, we aim to provide a clearer understanding of the dataset used in our study and highlight the robustness of our approach across varying levels of data availability. Hope the explanation clarifies the confusion and alleviates the reviewer’s concern.

  1. The authors described your workflow, but did not explain why you can get better results with a visual converter (ViT) model instead of a traditional CNN backbone. In other words, the authors only described how the model works, but not why it works.

Response: Thank you for the reviewer’s suggestion. We rewrite section 2.3 for more description about the MAE. We gave a detailed introduction about the mechanism of the MAE (page 5, lines 120-138) and the reason that the ViT was chosen as the backbone (page 6, lines 139-144). We hope the revised section addresses the reviewer’s concern.

Reviewer 3 Report

The manuscript is well organized. I do not have any technical comments for the authors. 

Author Response

Response: We really appreciate the reviewer’s positive comments and encouragement.

Round 2

Reviewer 1 Report

I carefully read the revised version of this manuscript. As can be understood, my questions are clarified, and previous issues are resolved. This manuscript is suitable for acceptance.